# Does the wind turbine wake follow the topography? - A multi-lidar study in complex terrain

Robert Menke[1], Nikola Vasiljević[1], Kurt S. Hansen[2], Andrea N. Hahmann[1], and Jakob Mann[1]

[1]Technical University of Denmark - DTU Wind Energy, Fredriksborgvej 399, Building 118, 4000 Roskilde, Denmark
[2]Technical University of Denmark - DTU Wind Energy, Nils Koppels Allé, Building 403, 2800 Kgs. Lyngby, Denmark

*Correspondence to:* Robert Menke (rmen@dtu.dk)

**Abstract.** The wake of a single wind turbine in complex terrain is analysed using measurements from lidars. A particular focus of this analysis is the wake deficit and propagation. Six scanning lidars, three short-range and three long-range WindScanners, were deployed during the *Perdigão 2015* measurement campaign, which took place at a double-ridge site in Portugal. Several scanning scenarios, including triple- and dual-Doppler scans, were designed to capture the wind turbine wake of a 2 MW turbine located on one of the ridges. Different wake displacements are categorised according to the time of the day. The results show a strong dependence of the vertical wake propagation on the atmospheric stability. When an atmospheric wave is observed under stable conditions, the wake follows the terrain down the ridge with a maximum inclination of -28°. During unstable conditions, the wake is advected upwards by up to 29° above the horizontal plane.

## 1 Introduction

Detailed knowledge about the behaviour of wind turbine wakes is essential for the design and the operation of wind farms. The wake is a region of reduced wind speed and increased turbulence in the downwind direction behind a wind turbine. It is caused by the extraction of kinetic energy from the wind field. As the wake propagates, it can affect other turbines in the downwind direction, which will produce less power due to the reduced kinetic energy of the flow (Barthelmie et al., 2007). Additionally, higher turbulence levels in the wake cause high fatigue loads, which in turn reduce the lifetime of wind turbines (Bustamante et al., 2015; Thomsen and Sørensen, 1999). Due to these facts and the trend to erect turbines in wind farms where turbines are relatively closely packed together, a detailed understanding of the wake propagation is imperative. Frequently applied computer models in the wind farm planning process include simplifications that are acceptable for sites with simple a orography. For example, the linearised model WAsP assumes that the wake follows the shape of the terrain in the downstream direction (Troen and Petersen, 1989). More sophisticated approaches like computational fluid dynamics (CFD) models and large eddy simulations (LES), do not constrain the wake propagation but cover prevailingly neutral atmospheric conditions apart from few academic studies (e.g. convective instabilities: Mirocha et al. (2014); stable conditions: Aitken et al. (2014); Bhaganagar and Debnath (2015); both convective and stable: Mirocha et al. (2015); Vollmer et al. (2016); Abkar and Porté-Agel (2015); Englberger and Dörnbrack (2017). To investigate to which degree these assumptions and limitations are applicable to real flow cases, we performed a full-scale experiment in complex terrain using wind lidars.

Wind lidars are available as continuous wave (CW) or pulsed systems. CW lidars are capable of providing high-frequency measurements of the line-of-sight (LOS) speed (up to 400 Hz, Vasiljević et al., 2017). However, at any given measurement rate CW lidars can only measure the LOS speed from a single range. Since the range determination is achieved by focusing the laser beam at a given point, the probe length is not constant with range. In fact, the probe length increases with the square of the range (Mikkelsen et al., 2008). This limits the maximum range of current CW lidars to about 150 m. Nevertheless, their capability of observing the wind field with a high frequency and relatively small probe lengths makes them ideal for studies of both mean and turbulent properties of the near wake in a region of up to two rotor diameters such as performed by Bingöl et al. (2010) and Trujillo et al. (2011). Unlike CW lidars, pulsed lidars can simultaneously measure LOS wind speeds from a number of ranges e.g., several hundred ranges in case of long-range WindScanner (Vasiljević et al., 2016), while their probe length is constant. The typical probe length is 30 m. Depending on their design, pulsed lidars can achieve a measurement range of up to 12 km (Krishnamurthy et al., 2013). However, pulsed lidars have a lower measurement rate than CW lidars, typically around 1 Hz. Due to the aforementioned characteristics, pulsed lidars are suitable for observations of the mean properties of the flow over larger distances and are used to study the far wake of wind turbines (i.e. Käsler et al., 2010; Iungo and Porté-Agel, 2013).

To the knowledge of the authors, the wake behaviour in highly complex terrain has not been investigated. All existing investigations of wakes with lidars took place in flat or moderately complex terrain and offshore (Bingöl et al., 2008; Rhodes and Lundquist, 2013; Smalikho et al., 2013; Aitken and Lundquist, 2014; Vollmer et al., 2015; Machefaux et al., 2016; Herges et al., 2017; Bodini et al., 2017). The present study focuses on wake measurements of a turbine located in highly complex terrain. During the *Perdigão 2015*, campaign which was a precursor for the larger *Perdigão 2017* campaign (Witze, 2017), the wake of a single wind turbine located on one of two parallel ridges was scanned. The terrain is characterised by patchy vegetation and an steep orography. For this purpose, six scanning lidars were deployed during a field campaign in the late spring and early summer of 2015, which is a part of a series of atmospheric experiments for wind energy (Mann et al., 2017). Three of the six lidars are long-range WindScanners (LRWS), pulsed Doppler lidars (Vasiljevic, 2014; Vasiljević et al., 2016) and three are short-range WindScanner (SRWS), CW wave lidars (Mikkelsen et al., 2008). The WindScanner system allows for the measurement of synchronised trajectories, thereby the retrieval of all three wind components is possible. The combination of both systems allows measuring the near and far wake, which is a novelty for wake measurements. The measurement analysis is concentrated on the vertical wake propagation and its relation to atmospheric stability. This dependency is of particular interest for turbines located in complex terrain in regions that are subjected to strong radiative forcings, e.g. the Mediterranean region. In these regions, the atmosphere is normally stable during the night and unstable during the day.

The paper is organised as follows. An overview of the designed scanning trajectories and a general description of the campaign is presented in section 2. In section 3 the available data and methods of their processing are introduced. Moreover, a new method to determine the inflow velocity in complex terrain is presented. Section 4 presents the results of the wake deficit analysis and wake propagation investigation. A summary and the conclusions of this work are given in the last section.

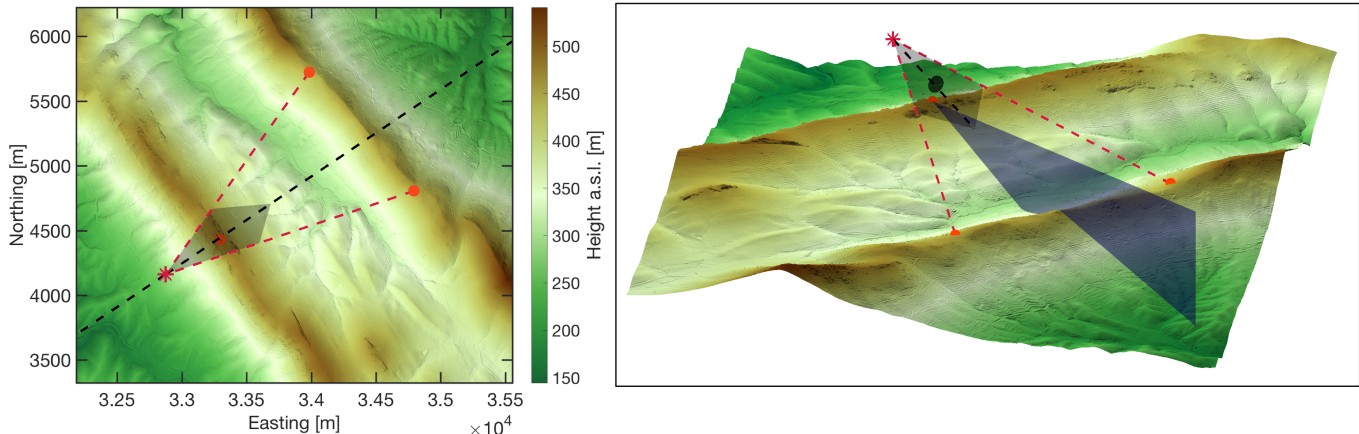

**Figure 1.** Overview of the terrain and scanning trajectories. Coordinates expressed in datum ETRS89/PTM06 and height above sea level. (red points: location of long-range WindScanners, blue plane: RHI scan, black circle: rotor plane, black plane: diamond scan plane, black dashed line: wind turbine transect or line of synchronised measurements in diamond scan, red dashed line: example of beams doing a measurement). Left: top view, Right: tilted view.

## 2 Field campaign overview

The *Perdigão 2015* field campaign took place during the months of May and June 2015. The site was selected to meet several scientific objectives. A detailed overview of the entire campaign is presented in Vasiljević et al. (2017). In this section, we give a summary of the campaign and site characteristics relevant for the measurements of the wind turbine wake.

### 2.1 Measurement site

The main characteristic of the site are two ridges that run in parallel for a distance of 2 km (Figure 1 and 2). The ridge to ridge distance is about 1.4 km and the valley bottom to ridge peak height equals to about 200 m. The ridges run from the southeast towards the northwest. The orography is generally lower to the northeast and southwest. The roughness of the terrain is determined by rocks and patchy vegetation of pine and eucalyptus trees. A former three-year assessment of the wind conditions by a 40 m tall meteorological mast located on the southwest ridge about 1 km in the southwest of the wind turbine shows two prevailing wind directions, northeast and southwest. These wind directions are perpendicular to the ridges. The wake measurements are obtained for a single Enercon E-82 2MW wind turbine located on the southwest ridge (see figure 2). This turbine has a hub height of 78 m and a rotor diameter of 82 m. It is designed as a low-wind speed turbine with a cut-in wind speed of 2 m/s. The terrain around the turbine drops of several tens of meters towards the southwest. Towards the northeast, the terrain slants down towards the valley after a smaller escarpment of about 20 meters height.

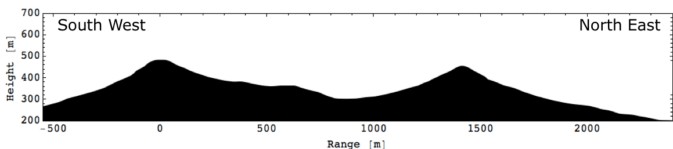

**Figure 2.** Wind turbine transect oriented $54°$ to North. The wind turbine is located at $0\,\text{m}$ range.

## 2.2   WindScanner measurements

Two WindScanner systems were deployed in the *Perdigão 2015* campaign, a short-range and a long-range WindScanner systems consisting of three short-range lidars and long-rage lidars, respectively. The short-range system was positioned close to the turbine, considering the shorter range and the probe volume size. For the SRWS the probe length increases quadratically with the measurement distance from about $0.2\,\text{m}$ at $10\,\text{m}$ to $40\,\text{m}$ at $150\,\text{m}$. Consequently, to keep the spatial averaging small, a measurement scenario is designed to measure the near wake of the turbine. The scenario consists of a vertical plane, perpendicular the turbine transect in the northeast of the wind turbine. Horizontally, the plane is centred at the turbine location extending $48\,\text{m}$ to each side. Vertically, the plane extends from $30\,\text{m}$ above the height of the turbine foundation to $128\,\text{m}$. The three SRWSs acquired synchronised measurements in this plane along 16 vertical lines (compare Figure 9b in Vasiljević et al., 2017). Accordingly, all three wind components ($u$, $v$, $w$) were measured. The 312 points were scanned 9 times during a 10-minute period, the system was started manually according to the present wind direction.

For the LRWSs, i.e. the pulsed lidars, the pulse length is the defining factor of the probe length. Considering relatively short distances of the test site a $200\,\text{ns}$ pulse was selected. With that pulse, the full-width half maximum of probe length is about $35\,\text{m}$ and the maximum measurement distance to about $3\,\text{km}$. For two scanners positioned on the Northeast ridge (Figure 1) a measurement trajectory was designed to measure both the turbine inflow and its wake in an almost horizontal plane. The plane is defined by the position of the two scanners and the wind turbine nacelle, resulting in an inclination of $4.7°$. The scanners were configured to acquire synchronised measurements in this plane along a line orientated $54°$ towards north with its centre at the wind turbine's nacelle. Due to the synchronous intersection of the two laser beams along this centre line instantaneous quasi-horizontal wind speeds and directions can be resolved. Measurement points are placed every $20\,\text{m}$, thus 50 in total, together with 2450 additional range gates placed next to this line, resulting in a diamond-shaped plane with the dimensions of $500\,\text{m} \times 750\,\text{m}$. The scanners measured this trajectory, named diamond scan, every half an hour for 10 minutes. A single scan took $25\,\text{s}$, resulting in 24 scans per 10-minute period. An RHI scan, a scan with constant azimuth angle and changing elevation angle, toward the Northeast along the wind turbine transect (Figure 1), was carried out by the third LRWS positioned close to the wind turbine. The scan covered elevation angles from $-12.5°$ to $20.5°$. In the plane of the RHI scan, a total of 12000 LOS velocities were acquired along 50 LOSs. Accordingly, this scan provides a field of LOS velocities. The LOS velocity is the flow component orientated along the lidar's laser beam. During certain periods that are analysed in detail in section 4.2, the wake could be captured with this scan and thus supplements the diamond scan in the vertical direction.

## 3 Data and Methods

### 3.1 Determination of inflow conditions

For analysing the wind turbine wake, detailed information about the inflow to the turbine is a requirement. This raises the question of how the inflow conditions can be described in complex terrain and particularly in this study. The IEC standard (IEC, 2005) provides that a reference measurement should be taken at a position $2-4$ rotor diameters upwind from the turbine. Such an approach is not appropriate in complex terrain. Influences of the orography, like the escarpments close to the turbine at the Perdigão site, cause significant perturbations to the flow field over small spatial scales.

Accordingly, inflow measurements should be taken close to the turbine, but at positions where the influence of the wind turbine itself to flow field is insignificant. To describe the turbines influence on the flow field, a model presented by Conway (1995) is chosen. The model shows analytically the influence of the rotor on the flow field under a specific loading of the turbine. Figure 3 shows the influence of the turbine in Perdigão on the free stream velocity ($U_\infty$) under maximum thrust. The maximal thrust is determined from the turbine's thrust curve and is observed below rated wind speed. We define $R$ as the radius of the wind turbine, $r$ the radial distance from the symmetry axis of the rotor and $x$ the distance perpendicular to the rotor. According to this model, measurement sections next to the rotor disk in the diamond scan are selected to represent the inflow conditions to the turbine (compare two grey sections in Figure 7a). The wind speed is averaged over two line segments, each 40 m long and positioned with a distance of 60 m to left and right of the wind turbine nacelle. The segments are always aligned perpendicular to the 10-minute averaged wind direction calculated over the whole diamond scan plane. Therefore, the position of these segments changes with the wind direction. In terms of the model by Conway, these segments are located in a range $r/R = 1.5$ to $r/R = 2.5$ and are at $x/R = 0$ and thus related to maximal wind speed changes ranging from 0.5% to 2.5%. However, since the blocking effect of the turbine at the position of the rotor is zero ($u/U_\infty = 1$), no effect on the selected segments is expected. For the case that the segments are not perfectly aligned with the rotor plane, the model shows that speed-up and slow-down effect cancel out in the case that the measurement volume is symmetric around the rotor, as it is in our case.

To validate this method the correlation with the wind turbine supervisory control and data acquisition (SCADA) data is investigated. The SCADA data is available as 10-minute averages for the period of the measurement campaign and includes the active power production, the nacelle direction and wind speed measured by the nacelle anemometer. From the measured power production in combination with the turbine's power curve, the rotor-equivalent wind speed $v_{\text{rotor}}$ is derived which represents the wind speed averaged over the entire rotor disk.

$$v_{\text{R}} = v_{\text{PC}}(P) \tag{1}$$

where $v_{\text{PC}}$ is interpolated from 1 m/s binned values of the turbine's power curve. The power curve is corrected for the Perdigão site following the IEC 61400-12 standard (IEC, 2005). An air density of $1.12\,\text{kg/m}^3$ is assumed resulting from an

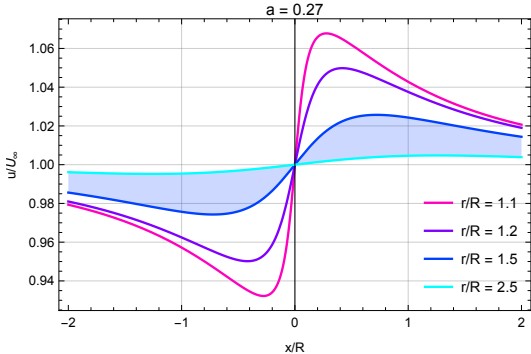

**Figure 3.** Solution of the actuator disk model by Conway (1995) for a induction factor of $a = 0.27$ and different ratios of the distance to the center line $r$ divided by the radius of the disk $R$. The shaded sector indicates the range in which the inflow measurement volume is located.

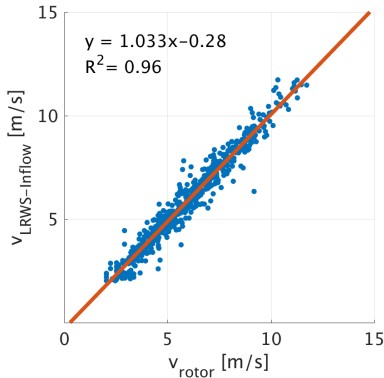

**Figure 4.** Correlation of the wind speeds derived from the diamond scan to the rotor equivalent wind speeds. The correlation is derived by 589 10-minute periods.

altitude of 562 m above mean sea level and an estimated average air temperature of 20C°. This method is only valid up to 12 m/s, the rated wind speed of the turbine since the power production is constant for higher wind speeds.

The linear correlation of $v_R$ and the wind speed acquired with the WindScanner system shows a good agreement in terms of a high coefficient of determination, $R^2 = 0.96$, and a slope close to one (figure 4). From the wind speed and direction measured by the diamond scan, a frequency distribution is calculated (figure 5). The rose shows the prevailing wind direction west-southwest and northeast, similar to a long-term assessment of the wind conditions at the site Vasiljević et al. (2017).

### 3.2 Data selection

During the campaign, 663 h of measurement data were collected by the WindScanner systems, 110 h with the SRWS system, which was in operation from May 8 until June 3 and 553 h with the LRWS which was in operation from May 19 until June 26. These numbers reduce by applying different filtering methods, as explained in Vasiljević et al. (2017). By considering the

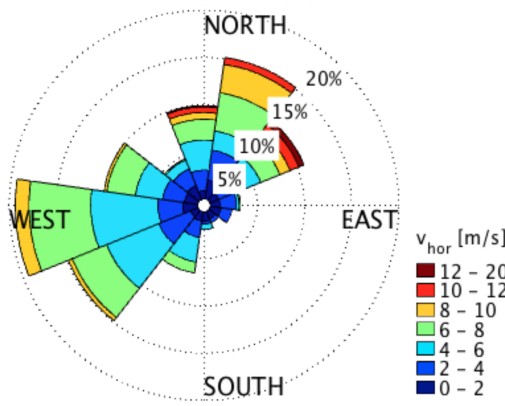

**Figure 5.** Sector wise frequency and velocity distribution derived from diamond scan measurements.

frequency distribution (figure 5) 10 h of LRWS measurement data is available in the northeastern sector. During these periods data from both the diamond scan and the RHI scan is available. Additionally, 5h of wake measurements are available from the SRWS system. Due to technical problems, no simultaneous measurements of the wake with the two WindScanner systems were collected (Vasiljević et al., 2017).

## 3.3 Mesoscale flow simulations

During the period of the campaign, strong radiative forcing was observed that caused significant diurnal changes of the air temperature at ground level. Due to the absence of meteorological stability measurements at the site during the 2015 campaign, simulations with the Weather Research and Forecasting (WRF) model are used to estimate the stability for the period of the campaign. The results are available covering the period between 1 May and 30 June 2015 and are produced with the WRF model version 3.6. For a description of the general WRF model, please refer to Skamarock et al. (2008).

The WRF model simulations are driven using ERA-Interim reanalysis datasets (Dee et al., 2011) using the method described in Hahmann et al. (2015) with the MYNN boundary layer parameterization. A total of three domains are used to downscale the flow field through one-way nesting, covering horizontal grid resolutions from 27 km to 3 km using a 1/3 ratio rule. Time series are extracted from the simulation results at the location of the wind turbine. Thermal stability is inferred from the Obukhov length $L_{mo}$ computed at the finest 3 km domain. At sites where stability measurements are available, the WRF model has shown good skill at simulating their climatology (Peña and Hahmann, 2012). However, the complexity of the local terrain is not considered, which can affect the interpretation of the stability length (Weigel et al., 2007; Rotach and Zardi, 2007).

The inner domain employs a 244×244×61 grid with the highest vertical level set at 50 hPa. The orography is generated using USGS 30" elevation data resampled according to grid resolution. Land-use categories for all domains are based on the CORINE Land Cover 2006 survey (CLC, 2006). For a detailed description of the simulation setup, please refer to Hahmann et al. (2017).

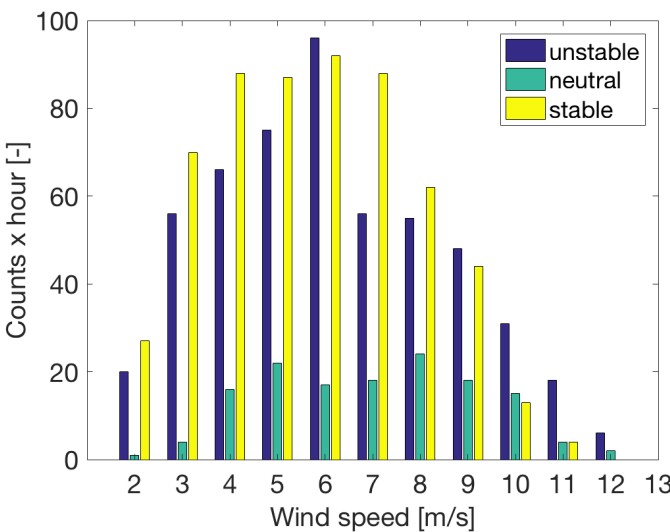

**Figure 6.** Stratification of the observations during the month May and June 2015. The wind speed is estimated from the rotor equivalent wind speed.

### 3.4 Atmospheric stratification at the wind turbine location

The WRF results show prevailingly stable atmospheric conditions during the night and unstable conditions during the day. Neutral conditions are present during transitions. We derive the stability from the Obukhov length using the following classification: stable $1/L_{mo} > 0.01$, neutral $0.01 > 1/L_{mo} > -0.01$, unstable $1/L_{mo} < -0.01$ similar to Muñoz-Esparza et al. (2012). The distribution of different stabilities over the rotor equivalent wind speed shows that both stable and unstable stratification are well represented, while the neutral stratification is infrequently represented (figure 6).

## 4 Results

### 4.1 Wake deficit and wake structure

Given the reference inflow wind speed obtained by the diamond scan, wake deficit profiles can be calculated in the measurement plane. These are based on horizontal wind speed retrievals from the dual-Doppler measurements. Deficit profiles are calculated at 1, 2 and 3 $D$ downstream of the wind turbine, where the velocity deficit in each point $i$ of a profile is defined as:

$$\Delta V_i = \left(1 - \frac{V_i}{V_{\text{inflow}}}\right) \tag{2}$$

An example of a 10-minute average period from 6:49 to 6:59 UTC, 7 June 2015 is shown in figure 7a. The profile extracted at 1 $D$ distance (figure 7b) shows a deficit with a local minimum in the wake centre, caused by a lower energy extraction around the nacelle. With increasing distance, this minimum disappears and the maximum deficit is decreasing from over 70% to 50%.

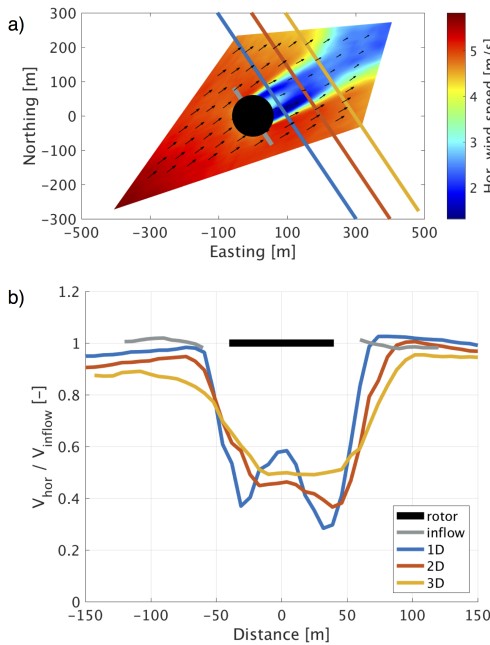

**Figure 7.** Wind turbine wake deficit observed with the diamond scan (10-minute averaged period measured on June 7, 2015 from 06:18 to 06:28): a) diamond scan with lines indication the position of deficit profiles; b) wake deficit profiles. The average tip speed ratio during this period is $\lambda = 6.7$ and the power coefficient is $c_p = 1.7 \cdot 10^{-4}$.

The averaged wake deficit for all available 10-minute periods in sectors to towards the northeast and southwest ($54° \pm 15°$ and $234° \pm 15°$) are calculated. The average maximum deficits are 53% (55%) at 1 D downstream for the northeastern (southwestern) sectors. For downstream distances of 2 D and 3 D, in some periods the wake moves above or below the diamond scan plane and therefore, the wake is not entirely captured by the diamond scan. This misalignment means that deficit profiles measured with the diamond scan may not capture the wake centre or the maximum wake deficit. Nevertheless, close to the turbine, the wake misalignment with the diamond scan is expected to be small, which allowed calculating an average deficit for specific sectors in a distance of one rotor diameter. Consequently, only selected periods can be investigated over the entire length of the diamond scan.

## 4.2 Wake Propagation

The wind turbine wake propagation is investigated for wind directions in a $30°$ sector centred around the wind turbine transect ($54°$). This sector is covered by both the diamond scan and the transect scan which makes a combined use of the scans possible. Moreover, this transect is aligned with the centre of the SR Wake Scan. The investigation of 10-minute averaged scans reveals significant vertical displacements of the wake. Considering both the state of the atmosphere captured by the transect scan and the WRF simulation results, four different cases are identified (see table 1).

**Table 1.** Wake categorisation according to stratification (the stratification is defined by the Obukhov length: stable $1/L_{mo} > 0.01$, neutral $0.01 > 1/L_{mo} > -0.01$, unstable $1/L_{mo} < -0.01$). Periods with atmospheric waves are identified by visual inspection of the transect scans.

| Case | Stratification | Wake advection |
|------|----------------|----------------|
| 1 | stable + atmospheric wave | downwards |
| 2 | stable | straight |
| 3 | neutral | straight |
| 4 | unstable | upwards |

Examples for each case are presented in figure 9. The first two cases are observed between sunset and sunrise during stable atmospheric conditions. These periods are categorised by distinct atmospheric layers of different wind speeds and directions. The transect scans for some nocturnal periods show a atmospheric wave pattern which interacts with the wake, displacing it downwards (case 1) in the direction of the leeward ridge slope, causing strong displacements of the wake centre. The wake
leaves the plane of the diamond scan after one or two rotor diameters downstream and continuous to propagates above the plane of the transect scan.

To relate the wake shapes in the diamond scan to a wake displacement, the cross-sectional area of an idealised wake and the diamond scan plane is geometrically modelled. The model, which contains code originally presented by Calinon (2009), represents the wake by a cone with an off-axis opening angle of $\Theta = 10°$ corresponding to the wake expansion determined at
2D distance for the period depicted in figure 9 (case 2). By varying the vertical angle of the cone, the length of the cross-section visible in the diamond scan is derived. The model shows that wakes are staying within the dimensions of the diamond scan plane for vertical displacement angles from $10°$ to $-19°$, see figure 8. For larger displacements only a cross-section of wakes is visible.

The categorisation and the geometrical model is applied to all periods for which the transect scan and the diamond scan are
available and the wind direction falls in the relevant sector. The wake length is determined from contour plots of the Diamond scan and transect scan determines the direction of vertical displacement. In total 21 periods are categorised in an upwards, downwards or straight wake propagation in relation to the atmospheric stability, see figure 10a. Two of the 21 periods show a wake centre displaced downwards, six a straight propagation of the wake and the remaining fourteen periods show an upwards moved wake centre. The wake propagates straight or downwards depending on the presence of a atmospheric wave during stable
conditions. Displacements upwards are observed mainly during unstable conditions. Inconsistencies of the categorisation and calculated stability occur in the evening times at around 18:00, where displacements downwards are observed and the WRF simulation shows neutral conditions. A reason for this can be that the WRF simulation has problems resolving the exact time of the transition from unstable to neutral conditions. For a clarification, direct measurements of the stability are necessary. However, the simulations agree in general with the categorisation and show an expected diurnal cycle for the period of the
campaign. Neutral conditions are expected at the transition from stable to unstable periods (or vice versa). The cross-sectional

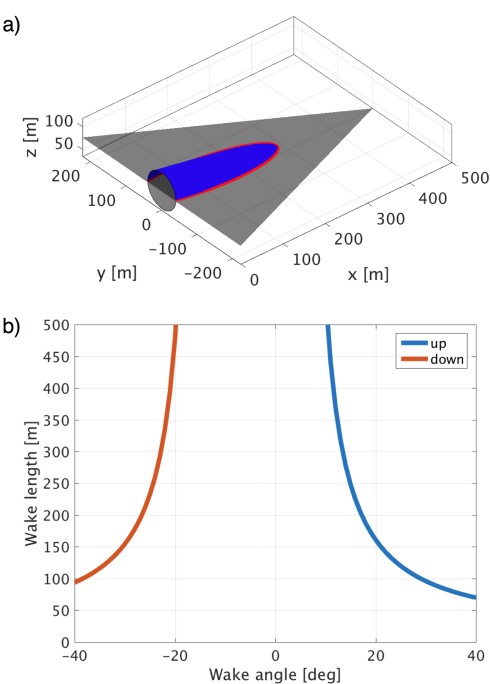

**Figure 8.** Wake cross section model: a) Model schematic for a negative vertical wake angle of -23° (black plane: one half of the diamond scan plane, blue plane: cone section representing the wake, red line: cross section, black circle: rotor plane); b) Derived length of the wake cross section in the diamond scan for different wake angles with a wake expansion angle of $\Theta = 10°$. The asymmetry of the model results from the negatively inclined scanning plane.

model shows wake angles from 21° to 29° during stable conditions and angles of up to $-28°$ in combination with the presence of an internal wave (Figure 10b).

Results of the of the short-range WindScanner scan in one rotor diameter distance indicate an equal behaviour of the wake propagation. Periods are observed where the wake centre is moved downwards or lifted upwards. In case of a downwards moved wake it is observed that there is no clear distinction between the wake and the flow field under the rotor (Figure 11 b). The upwards moved case, on the other hand, shows clearly the flow passing between ground and rotor disc. The short-range could provide only a small amount of measurements, due to a small scanning plan, which only allowed to capture the wake for a narrow sector of wind directions and technical problems with the system.

## 5    Summary and conclusions

Measurement data from six scanning lidars have been analysed with the purpose of describing the wind turbine wake propagation in complex terrain. The data, recorded in the summer of 2015, was collected at a site with highly complex orography in the centre of Portugal, near the town of Perdigão. The analysis focuses on the vertical wake propagation under different atmo-

spheric stability conditions, whose influence is generally ignored in computational flow simulations, namely those performed for wind-farm layout-design. The combined analysis of a horizontal scanning plane, centred around the wind turbine, and a transect scan of radial wind speeds in that sector revealed strong vertical wake angles. In total, four different wake cases are identified that show, among horizontal wake propagation, strong displacements upwards and downwards of the wake during stable and unstable atmospheric conditions. These displacements are caused by the flow field that is influenced by the atmospheric stability and the orographic structure of the terrain around the turbine. By including data from a WRF simulation that provides estimates of the atmospheric stability in terms of the Obukhov length, a correlation between wake angles and stability emerges. During unstable conditions, the wake is lofted up by up to $29°$. During stably-stratified conditions, wakes follow the terrain, resulting in downward wake angles reaching $-28°$. Under neutral conditions, the wake propagates approximately horizontally despite the complex orography. Wake deficits of 70% are measured at $1\,D$ distance, decreasing to 50% in $3\,D$ distance. Differences in wake characteristics observed with the scanning trajectories of this study arise from a combination of angles between wake and scanning plane, atmospheric conditions, and the operational state of the turbine. With the present dataset, it is not possible to isolate the effect of these factors.

This study shows that wake propagation is highly complex and strongly depends on the stability conditions and the terrain orography. The question "Does the wake follow the terrain?" is answered: in some stable conditions the wake does follow the terrain, but in the majority of cases observed here, the wake does not follow the terrain. However, the wake could not be observed over the full vertical range with the scanning trajectories of this campaign as the designed trajectories were addressing several scientific objectives at the same time (see Vasiljević et al., 2017). For future investigations of the wind turbine wake propagation in complex terrain, it can be recommended to design scanning trajectories in a way that they cover a wider vertical range and to focus optimally only on a single scientific objective. Furthermore, a nacelle mounted spinner lidar (Mikkelsen et al., 2010) could provide more detailed information about the inflow to the turbine, as this investigation showed that it is challenging to describe the inflow in complex terrain.

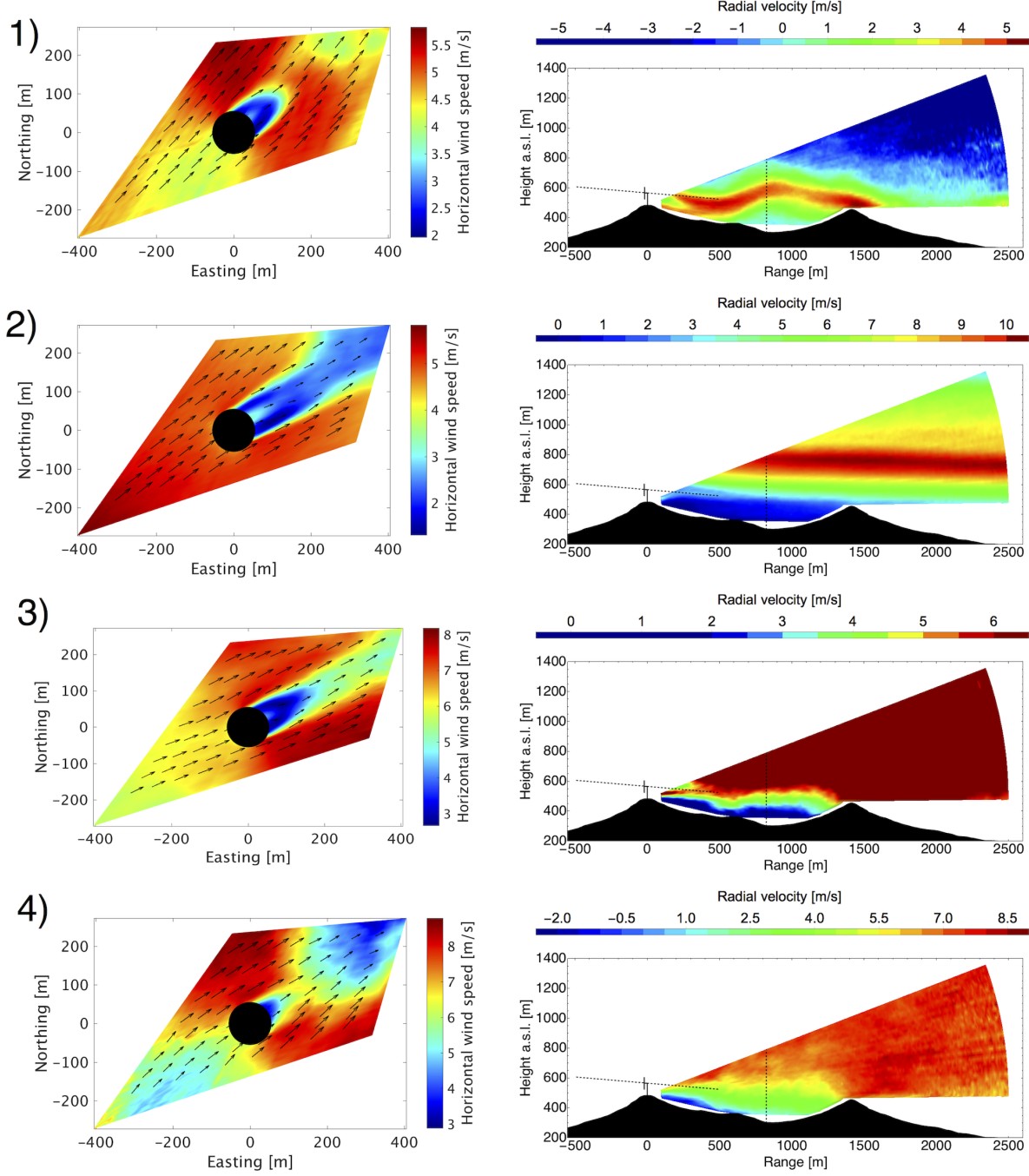

**Figure 9.** Observed wake cases with the diamond scan (left) and transect scan (right). 1) downwards displaced wake, stable atmospheric conditions + atmospheric wave, 2) no displacement, stable atmos. conditions, 3) no displacement, neutral atmos. conditions 4) upwards displaced wake, unstable atmos. conditions. (measurements of example periods taken: 1) 02:52 – 03:02 UTC 9 June 2015, 2) 06:18 – 06:28 UTC 7 June 2015, 3) 22:20 – 22:30 UTC 10 June 2015, 4) 11:16 – 11:26 UTC 10 June 2015 )

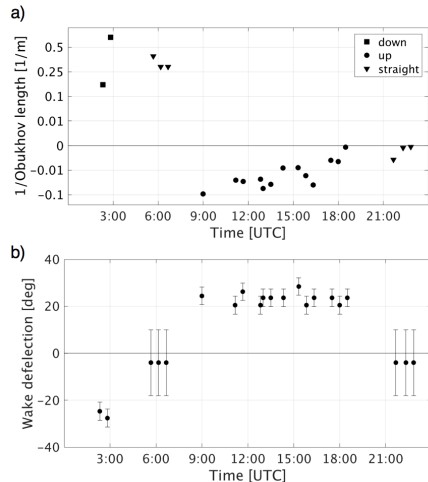

**Figure 10.** Wake displacements as a function of the time of the day: a) direction of displacement relative to one over the Obukhov length; b) wake deflection angles according to the cross sectional model. The error is indicating the sensitivity to the wake expansion angle for changes of $\Delta\Theta=\pm2°$ and for periods with a straight propagation the range in which wakes are fully covered by the diamond scan.

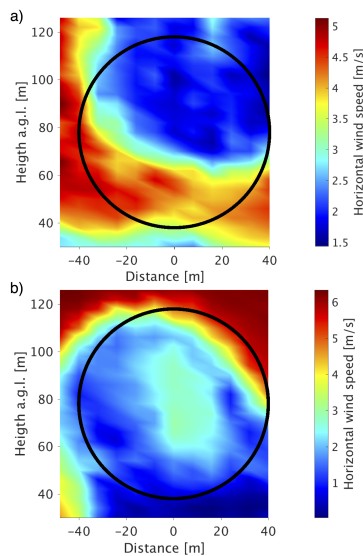

**Figure 11.** Short-range WindScanner wake measurements a) upwards moved wake (10-minute period from 19:20 – 19:30 UTC 11 May 2015) and b) downwards moved wake (10-minute period form 00:50 – 01:00 UTC 12 May 2015), further examples are presented in Hansen et al. (2016). Black circle: projected rotor disc.

*Author contributions.* R. Menke analysed the data with support from N. Vasiljević and K. Hansen. R. Menke developed the method used to determine the inflow to the wind turbine and characterised the wake behaviour with the help and supervision of J. Mann and K. Hansen. N. Vasiljević and J. Mann designed the field experiment, with feedback from K. Hansen, and executed it. A. N. Hahmann performed the WRF simulation. R. Menke wrote the main body of manuscript with inputs from all authors. All authors contributed with critical feedback to this
research.

*Competing interests.* The authors declare that they have no conflict of interest.

*Acknowledgements.* All authors thank Carlos Rodrigues for his great inputs to this research. We also acknowledge the work of José Laginha Palma and Jose Carlos Matos. Their great contribution during all phases of the field campaign helped to make the campaign a success. We also thank Mike Courtney for his work in the planning and design process of the campaign. We would like to thank Nikolas Angelou for
the operation of the short-range WindScanner system and his help with the data processing, and Guillaume Lea, who helped to process the data of the long-range WindScanner system. Furthermore, the authors acknowledge the FarmOpt project for the financial support for the *Perdigão 2015* field campaign. FarmOpt (http://energiforskning.dk/en/projects/detail?program=All&teknologi=All&field_bevillingsaar_ value=&start=&slut=&field_status_value=All&keyword=FarmOpt&page=0) was funded by the Danish Energy Technology Development and Demonstration Program (EUDP), project no. 64013-0405. We thank the Danish Energy Agency for funding through the New European
Wind Atlas project.

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
