# Peer review of "Does the wind turbine wake follow the topography? - A multi-lidar study in complex terrain"

_Wind Energy Science, 2018_

## Referee Comment (RC1) · M. Guala (Referee) · 15 Apr 2018

The manuscript is interesting and poses a relevant question for wind power plant layout in complex terrain. Authors employed state of the art field measuring technology, which makes the dataset unique and valuable.

There are however a few issues that I would like to point out, which could lead to an improvement of this work and perhaps a broader impact in the community.

1) I have some reservation on the atmospheric stability assessment: $|z/L| <0.01$ is a very strict condition for the neutral regime, rarely observed from micrometeorological data from sonic anemometers. Based on Fig 6 it seems that it occurs quite frequently. I am wondering how accurate is the estimate of the turbulent heat flux and how far from

the surface (the actual z) is the estimate referring to.

2) More importantly the Monin Obukhov similarity assumes a logarithmic region where the mean velocity profile is distorted by the thermal stability effect. In complex terrain the contributions to mechanical production of turbulent kinetic energy may be more complicated as compared to the $u*\hat{}3/kz$ term that is likely employed here. The authors should provide the definition of L and discuss how they account for the non-flat topography and for the orientation of the reference system with respect to the mean incoming wind (likely non flat and to some extent following the terrain).

3) Fig 7: the wake deficit depends on the turbine operating conditions: it would be relevant to provide the tip speed ratio and the power coefficient for the wake plotted in Fig 7b (at least the 10min corresponding averaged value).

4) Despite of many hours of measurements, the most interesting figures show results from quasi-instantaneous measurements. I wonder if it is possible to use conditional averages or two point correlation to support the conclusion with statistics instead of single realization. Perhaps, the wind tunnel work by KB Howard, LP Chamorro, and M Guala "comparative analysis on the response of a wind-turbine model to atmospheric and terrain effects" Boundary-layer meteorology 158 (2), 229-255, 2015 may offer some ideas.

5) Fig 10b: how is the wake deflection angle estimated? within a range of x/D? based on the velocity contour, a velocity minima envelope? Please clarify

6) For non LiDAR experts, perhaps the definition of radial velocity should be provided. Some of the velocity contour distribution with height presented in fig 9b are prone to be misinterpreted without a proper definition.

Best, Michele Guala

---

## Referee Comment (RC2) · Anonymous Referee #2 · 16 May 2018

General comment: The manuscript describing wind turbine wake measurements is well written, of good scientific value and probably the first publication describing turbine wake measurements in complex terrain. The biggest challenge with this manuscript is the very short period of wake observations. Due to this, making any definite conclusions is not possible. However, this manuscript still deserves publication after a few minor revisions.

Specific comments: Since the scan strategy is not suited to study more than a couple of rotor diameters of the wake extent, it would be useful to go into the details of the deficit magnitude. For example, does the deficit magnitude vary as a function of turbine operation and stability?

From Figure 9, I see that the deficit minimum at the wake center seems to exhibit

different characteristics for the four different cases. For example, the length of the minimum at the center seems to vary for the different conditions. Is this a function of the stability or turbine operation? Do these differences show up in the other cases?

There are a few places in the manuscript that could use some grammar corrections. I have identified some of them. However, I am not proficient in English myself and therefore it might be a good idea to get this manuscript proof read by an expert rather than rely on my suggestions.

1) Page 1, line 14: "cause" and not "are causing"

2) Page 1, line 20: "cover" and not "are covering"

3) Page 5, line 7: "cause" and not "are causing"

---

## Author Comment (AC1) · 5 Jul 2018

**Response to reviewer 1 (Michele Guala)**

We thank Michele Guala for his critical feedback which helped us to improve this manuscript. Below we answer his specific comments in detail:

**Comment 1**
*I have some reservation on the atmospheric stability assessment: |z/L| <0.01 is a very strict condition for the neutral regime, rarely observed from micrometeorological data from sonic anemometers. Based on Fig 6 it seems that it occurs quite frequently. I am wondering how accurate is the estimate of the turbulent heat flux and how far from the*

*surface (the actual z) is the estimate referring to.*

The |z/L| used in the stability assessment is derived from the RMOL variable in the WRF output, which is from the formulation in the MYNN2 PBL and surface layer scheme (Nakanishi and Niino 2006) and is computed at the model surface (10m). As mentioned in the manuscript, there is some evidence that the WRF model has shown good skill at simulating their climatology (see e.g. Fig 6 in Draxl et al, 2014). The limit of |z/L| < 0.01 is also used by Muñoz-Esparza et al. (2012) who suggests a limit of |L| > 1000.

**Comment 2**

*More importantly the Monin Obukhov similarity assumes a logarithmic region where the mean velocity profile is distorted by the thermal stability effect. In complex terrain the contributions to mechanical production of turbulent kinetic energy may be more compli-cated as compared to the $u_\star^3/kz$ term that is likely employed here. The authors should provide the definition of L and discuss how they account for the non-flat topography and for the orientation of the reference system with respect to the mean incoming wind (likely non flat and to some extent following the terrain).*

We are aware that the Monin Obukhov theory is meaningless when calculated for flow over complex terrain from mast measurements. However, the simulation results of the WRF model employed in this study are based on a 3 km x 3 km grid of the smallest domain. Terrain features which are small compared to the model resolution, like the Perdigão ridges, are thus not visible to the model and are only represented by an increased surface roughness. The model sees only a smoothed surface which makes it possible to derive the Obukhov length from the simulation. Confidence in the results is gained by a comparison of the WRF results with the diurnal cycle (Figure 1). The distribution shows a clear separation of the results in a day and night time regime. This distribution corresponds well with observations made by us in the field during the campaign. The period of the campaign was generally very hot and dry with maximum

temperatures above 40°C.

**Comment 3**
*Fig 7: the wake deficit depends on the turbine operating conditions: it would be relevant to provide the tip speed ratio and the power coefficient for the wake plotted in Fig 7b (at least the 10min corresponding averaged value).*

We added the tip speed ratio and power coefficient based on the 10 minute averages from the SCADA system of the turbine to the manuscript.

**Comment4**
*Despite of many hours of measurements, the most interesting figures show results from quasi-instantaneous measurements. I wonder if it is possible to use conditional averages or two point correlation to support the conclusion with statistics instead of single realization. Perhaps, the wind tunnel work by KB Howard, LP Chamorro, and M Guala "comparative analysis on the response of a wind-turbine model to atmospheric and terrain effects" Boundary-layer meteorology 158 (2), 229-255, 2015 may offer some ideas*

We will consider employing the two-point correlation technique for further work that will be done also in combination with a new dataset that was measured during the Perdigão 2017 campaign at the same measurement site.

**Comment 5**
*Fig 10b: how is the wake deflection angle estimated? within a range of x/D? based on the velocity contour, a velocity minima envelope? Please clarify*

The length of the wake in the Diamond scan is estimated by the velocity contour. This information has been added to the manuscript.

**Comment 6**

*For non LiDAR experts, perhaps the definition of radial velocity should be provided. Some of the velocity contour distribution with height presented in fig 9b are prone to be misinterpreted without a proper definition.*

The definition of radial velocity has been included in the manuscript and the description of the RHI scanning trajectory has been advanced.

**References**

Draxl, C., Hahmann, A. N., Peña, A. and Giebel, G.: Evaluating winds and vertical wind shear from Weather Research and Forecasting model forecasts using seven planetary boundary layer schemes: Evaluation of wind shear in the WRF model, Wind Energy, 17(1), 39–55, doi:10.1002/we.1555, 2014.

Muñoz-Esparza, D., Cañadillas, B., Neumann, T. and van Beeck, J.: Turbulent fluxes, stability and shear in the offshore environment: Mesoscale modelling and field observations at FINO1, Journal of Renewable and Sustainable Energy, 4(6), 063136, doi:10.1063/1.4769201, 2012.

[Figure]

**Fig. 1.** Diurnal cycle of the Obukhov length for the period of the measurement campaign.

---

## Author Comment (AC2) · 5 Jul 2018

**Response to reviewer 2**

We thank the anonymous reviewer for her or his feedback and comments. Below we answer the specific comments in detail:

**Comment 1**

*Since the scan strategy is not suited to study more than a couple of rotor diameters of the wake extent, it would be useful to go into the details of the deficit magnitude. For example, does the deficit magnitude vary as a function of turbine operation and stability?*

It would be interesting to analyze the relation of wake deficit to other parameters. However, we cannot determine where exactly the scanning trajectory (Diamond Scan) cuts the wake makes it impossible to determine the absolute velocity deficit. This fact prohibits to set the deficits derived from the Diamond scan in relation to the atmospheric stability or turbine operational parameters. We are addressing this issue in section 4.1: "Such a misalignment causes that deficits profiles measured with the Diamond scan are not necessarily located in the wake center. Nevertheless, in short distance to the turbine, the wake misalignment with the Diamond Scan is expected to be small, which allowed calculating an average deficit for specific sectors in a distance of one rotor diameter. Consequently only selected periods can be investigated over the entire length of the Diamond Scan."

**Comment 2**
*From Figure 9, I see that the deficit minimum at the wake center seems to exhibit different characteristics for the four different cases. For example, the length of the minimum at the center seems to vary for the different conditions. Is this a function of the stability or turbine operation? Do these differences show up in the other cases?*

The operational state of the turbine and atmospheric conditions have undoubtedly an influence on the wake characteristics. The wake measurements shown in figure 9 are used to illustrate the strong vertical wake displacements. Differences in the wake characteristic observed with the Diamond Scan are a combination of different angles between wake and scanning plane, atmospheric conditions and the operational state of the turbine. Unfortunately, it is not possible with the available measurements to isolate one of these factors.

**Comment 3**
*There are a few places in the manuscript that could use some grammar corrections. I have identified some of them. However, I am not proficient in English myself and*

*therefore it might be a good idea to get this manuscript proof read by an expert rather than rely on my suggestions.*
*1) Page 1, line 14: "cause" and not "are causing"*
*2) Page 1, line 20: "cover" and not "are covering"*
*3) Page 5, line 7: "cause" and not "are causing"*

Thanks for these corrections. We carefully read the manuscript again to improve the presentation of this research.

---

## Author Response (AR2)

**Response to associated editor revisions:**

Dear Julie Lundquist,

we thank you for reviewing this manuscript, it helped us a lot to improve this work. Please, find below a point-by-point response to your suggested revisions.

*1. Page 7, line 7: insert "during the 2015 campaign" to make clear that the lack of stability measurements was only during the 2015 campaign and not during subsequent campaigns to avoid confusion.*

We included the reference "during the 2015 campaign" in the manuscript.

*2. Page 7, line 15: replace "finer" with "finest"*

Replaced.

*3. Page 7, line 16: eliminate "the few" because there are tens of studies comparing WRF simulations with stability measurements from field campaigns.*

Thanks for the remark, we removed "the few" from the manuscript.

*4. Page 7, near line 15: please acknowledge reviewer 1's comment regarding the meaning of L_mo in complex terrain by including a sentence clearly stating that the 3-km resolution runs do not consider the complexity of the local terrain, which can affect interpretation of stability length, and cite the appropriate literature (perhaps https://rmets.onlinelibrary.wiley.com/doi/abs/10.1002/qj.71 and references cited therein).*

Following sentence was added to the manuscript to address this point: "However, the complexity of the local terrain is not considered, which can affect the interpretation of the stability length (Weigel et al., 2007; Rotach and Zardi, 2007)."

*5. Throughout: why are "Diamond Scan" and "Transect Scan" capitalized (and inconsistently).*

The "Diamond Scan" and "Transect Scan" were considered as proper names for the unique scanning trajectories. However, we agree that this can be irritating for the reader and switched to lower case spelling throughout the manuscript.

*6. Page 9: very confusing as written. Try: "The average maximum deficits are 53% (55%) at 1 D downstream for the northeastern (southwestern) sectors. For downstream distances of 2 D and 3 D, in some periods the wake moves above or below the diamond scan plane and therefore, the wake is not entirely captured by the diamond scan. This misalignment means that deficit profiles measured with the diamond scan may not capture the wake center or the maximum wake deficit."*

Thanks for the suggested reformulation, we adopted it.

*7. Page 12, line 9: have you shown that gravity waves are completely responsible for the downward trajectory of the wake in stable conditions? (I think not: there is only a casual visual inspection to identify gravity waves, without the necessary analysis to support their identification as gravity waves as opposed to any other waves.) Smoke plumes follow terrain in stable conditions even without gravity waves. I recommend omitting the reference to gravity waves, so "During stably stratified conditions, wakes follow the terrain, resulting in ....."*

We agree that we could not show that gravity waves are completely responsible for the wake behavior in some cases. The reference to gravity waves in the conclusion has been removed.

*8. Page 12: somewhere in the conclusions, the authors should repeat their response to reviewer 2 comment 2, that "The operational state of the turbine undoubtedly affects wake characteristics. These operational states may be defined by blade pitch angle (https://www.wind-energ-sci.net/1/221/2016/wes-1-221-2016.html), yaw angle error (http://iopscience.iop.org/article/10.1088/1742-6596/524/1/012002/pdf)) , among other operational characteristics. Differences in wake characteristics observed with the diamond scan arise from a combination of angles between wake and scanning plane, atmospheric conditions, and the operational state of the turbine. Unfortunately, it is not possible with this dataset to isolate one of these factors."*

We added the following sentence to the conclusion: "Differences in wake characteristics observed with the scanning trajectories of this study arise from a combination of angles between wake and scanning plane, atmospheric conditions, and the operational state of the turbine. With the present dataset, it is not possible to isolate the effect of these factors."

*Minor: I also identified several English grammar issues, so please carefully review the entire manuscript again. My careful review of the conclusion section found the following:*
*1. Page 12: line 3: comma after "total"*
*2. Page 12: line 8: replace "is identified" with "emerges"*
*3. Page 12: line 8: "lofted" instead of "lifted"?*
*4. Page 12, line 8: comma after "conditions" and eliminate "the"*
*5. Page 12, line 9: replace "is dependent on" with "depends on"*
*6. Page 12, line 10: comma after conditions, omit "of" after despite*
*7. Page 12, line 11: replace "in" with "at"*
*8. Page 12, line 12: "that wake propagation" instead of "that the wake propagation"; "depends" instead of "depended"*

[revised manuscript text omitted]

---

## Author Response (AR3)

**Response to associated editor revisions:**

Hi Julie,

thanks for point out the reference to gravity waves in the rest of the manuscript. I changed the reference throughout to "atmospheric waves".

Thanks,
Robert

[revised manuscript text omitted]